# Stress Distribution on Spinal Cord According to Type of Laminectomy for Large Focal Cervical Ossification of Posterior Longitudinal Ligament Based on Finite Element Method

**DOI:** 10.3390/bioengineering9100519

**Published:** 2022-10-02

**Authors:** On Sim, Dongman Ryu, Junghwan Lee, Chiseung Lee

**Affiliations:** 1Department of Biomedical Engineering, Graduate School, Pusan National University, Busan 46241, Korea; 2Medical Research Institute, Pusan National University, Busan 46241, Korea; 3Department of Neurosurgery, School of Medicine, Pusan National University, Busan 46241, Korea; 4Department of Convergence Medicine and Biomedical Engineering, School of Medicine, Pusan National University, Busan 46241, Korea; 5Biomedical Research Institute, Pusan National University Hospital, Busan 49241, Korea

**Keywords:** ossification of posterior longitudinal ligament, finite element analysis, laminectomy, spinal cord, decompression, motion

## Abstract

Most studies on the ossification of the posterior longitudinal ligament (OPLL) using the finite element method were conducted in the neutral state, and the resulting decompression was judged to be good. As these studies do not reflect the actual behavior of the cervical spine, this study conducted an analysis in the neutral state and a biomechanical analysis during flexion and extension behaviors. After validation via the construction of an intact cervical spine model, the focal OPLL model was inserted into the C4–C5 segment and a simulation was performed. The neutral state was shown by applying a fixed condition to the lower part of the T1 and *Y*-axis fixed condition of the spinal cord and simulating spinal cord compression with OPLL. For flexion and extension simulation, a ±30-degree displacement was additionally applied to the top of the C2 dens. Accordingly, it was confirmed that spinal cord decompression did not work well during the flexion and extension behaviors, but rather increased. Thus, if patients with focal OPLL inevitably need to undergo posterior decompression, additional surgery using an anterior approach should be considered.

## 1. Introduction

The human spine supports the body’s weight, aids in movement, and plays an important role in protecting the spinal cord. Among the vertebrae, the cervical spine is the most frequently injured part, and cervical spine injury can be life-threatening [1,2,3]. In addition, the cervical vertebrae have a wider range of behaviors than other vertebrae. Thus, ossification of the posterior longitudinal ligament (OPLL) in this area has a greater effect on the spinal cord; therefore, special attention is required [4].

OPLL is in the spinal canal, which is located at the back of the vertebral body. In this condition, the posterior ligament, one of the ligaments connecting the top to the bottom of the vertebrae, becomes abnormally stiff and enlarges. In other words, the ligaments that help in the normal movement of the spine, are ossified and enlarged against the spinal cord and cause neurological disorders [5].

Neural decompression is the only treatment available for OPLL. Some methods involve the removal of ossified ligaments, while other methods include simple decompression without removal of such ligaments. Anterior decompression is generally required to remove ossified ligaments, especially the focal type. Ossified ligament removal is a fundamental treatment but has a high risk of fatal complications due to dura mater or spinal cord damage during surgery, and it is difficult to treat multi-segmental lesions. The anterior approach is preferred in patients with kyphosis with OPLL at 1 level or 2 level. Meanwhile, posterior decompression is generally preferred for simple neural decompression. With this procedure, the incidence of fatal complications is low, and decompression of several segments is easy [6,7,8]. The posterior approach is preferred for multiple lesions with cervical lordosis. On the other hand, in the case of one level and lordosis, both anterior and posterior approaches can be considered [9]. Although this posterior approach has many advantages, biomechanical problems may occur when the cervical vertebrae are moved, depending on the type of OPLL remaining after surgery. This may be attributed to the fact that in the posterior approach, ossified ligaments are not removed.

However, the studies that have been conducted thus far are limited to the stresses that act on the spinal cord and nerve roots in the neutral state [10]. Therefore, in this study, OPLL and laminectomy models were constructed; flexion and extension behaviors were simulated, and the results were analyzed.

## 2. Materials and Methods

### 2.1. Fabrication and Validation of the Intact Cervical Spine Model

A three-dimensional (3D) intact cervical spine model was constructed to simulate OPLL and apply laminectomy. Generally, the cervical spine comprises various hard/soft tissues. To construct a standardized model, we developed a cervical spine model with an average lordotic value based on clinical knowledge and clinical paper data [11,12,13,14,15,16,17]. The intact cervical spine model consists of the cortical bone, cancellous bone, intervertebral disc (IVD), anterior longitudinal ligament (ALL), posterior longitudinal ligament (PLL), capsular ligament (CL), ligamentum flavum (LF), interspinous ligament (ISL), supraspinous ligament (SSL), and spinal cord [Figure 1].

The material properties and cross-sectional areas were based on previous clinical data and consisted of tension-only beams for the ALL, PLL, ISL, and SSL (Table 1 and Table 2). In the case of IVD, Mooney–Rivlin coefficients were applied as a hyperelastic model, and the remaining materials were configured as a linear elastic isotropic model.

Flexion and extension simulation were performed by applying the physical properties of each tissue to the intact cervical spine model and constructing a finite element model to apply ±1 Nm to the top of the C2 dens and a fixed condition to the lower part of T1. Since it was verified with the experimental data value of Panjabi, the same ±1 Nm as Panjabi was applied. The simulations showed that the range of motion (ROM) of each segment was comparable to the data obtained by Panjabi [18].

### 2.2. Fabrication of the OPLL Cervical Spine and Laminectomy Models

The OPLL model was constructed with three types of covered disc, covered vertebra, and connected vertebra, which showed a large number of frequencies [19] [Figure 2]. Type 1 OPLL is covered disc, type 2 OPLL is covered vertebra, and type 3 OPLL is connected vertebra.

Generally, posterior decompression aims to achieve decompression by securing space. Therefore, in this study, laminectomy was simulated when C4 and C5 lamina were removed [Figure 3].

### 2.3. Fabrication of the Finite Element Model and Simulation

A finite element model including a total of 12 cases was constructed based on three cases of the OPLL cervical spine model and three types of laminectomies. Finite element model’s mesh element types are on Table 3 and mesh quality average was 0.77, skewness average was 0.34, orthogonal quality average was 0.69. The simulation was performed by applying a fixed condition to the lower part of the T1 and *Y*-axis fixed condition of the lower part of the spinal cord. Thereafter, neutral simulation was performed, in which the OPLL model was placed in the cervical vertebrae in the neutral state and compressed the spinal cord while moving toward the spinal cord. The OPLL growth gradually progressed. The finite element model for flexion and extension simulation was constructed by extracting the spinal cord shape in 12 cases, which were changed by simulation of the neutral state. For flexion and extension simulation, ±30-degree remote displacement was additionally applied to the top of the C2 dens.

## 3. Results

### 3.1. Model Validation

For verification, the ROM of each segment was measured when flexion and extension were simulated using the remote point set in the ANSYS program and the flexible rotation probe function at the center point of each upper vertebral body [20,21,22,23]. The results of the intact cervical spine model were confirmed to be within the range of the cadaveric experimental data reported by Panjabi. The ROM in the finite element model in this study was also within the value reported by Panjabi, who performed the experiment on a human body; the finite element model does not match the shape with the experiment, but it is considered suitable for simulating the behavior of the human body [Figure 4 and Figure 5].

### 3.2. Neutral Laminectomy Simulation

As a result of neutral laminectomy simulation, in type 1 OPLL, the spinal cord moved 0.6464 mm (C4 direction) posteriorly during C4 laminectomy, 1.1641 mm (C5 direction) posteriorly during C5 laminectomy. It moved 0.4882 mm (C4 direction), 1.1596 mm (C5 direction) posteriorly during C4–C5 laminectomy.

In type 2 OPLL, the spinal cord moved 0.6629 mm (C4 direction) posteriorly during C4 laminectomy, 0.9934 mm (C5 direction) posteriorly during C5 laminectomy. It moved 0.5767 mm (C4 direction), 0.9273 mm (C5 direction) posteriorly during C4–C5 laminectomy.

In type 3 OPLL, the spinal cord moved 0.6255 mm (C4 direction) posteriorly during C4 laminectomy, 1.1541 mm (C5 direction) posteriorly during C5 laminectomy. It moved 0.5424 mm (C4 direction), 1.1524 mm (C5 direction) posteriorly during C4–C5 laminectomy.

The stresses and shapes acting on the spinal cord when laminectomy is applied to each type of OPLL are shown below [Figure 6, Figure 7 and Figure 8].

The results of the neutral simulation and decompression are shown in Figure 9 and Table 4, Table 5 and Table 6.

The % expressed below is the decompression rate when each laminectomy is applied based on the non-surgery model for each type of OPLL. The −(minus) value indicates that the stress acting on the spinal cord increased.

The decompression rate of the maximum principal stress value in, the case of type 1 OPLL was 28.40% for C4 laminectomy, 35.26% for C5 laminectomy, and 60.37% for C4–C5 laminectomy. In type 2 OPLL, the decompression rate was 21.93% for C4 laminectomy, 43.63% for C5 laminectomy, and 63.32% for C4–C5 laminectomy. In type 3 OPLL, the decompression rate was 27.12% for C4 laminectomy, 37.38% for C5 laminectomy, and 60.54% for C4–C5 laminectomy.

The decompression rate of the minimum principal stress value, in the case of type 1 OPLL was 21.60% for C4 laminectomy, 18.78% for C5 laminectomy, and 62.07% for C4–C5 laminectomy. In type 2 OPLL, the decompression rate was 0.85% for C4 laminectomy, 51.59% for C5 laminectomy, and 60.69% for C4–C5 laminectomy. In type 3 OPLL, the decompression rate was 22.65% for C4 laminectomy, 12.61% for C5 laminectomy, and 58.26% for C4–C5 laminectomy.

The decompression rate of the maximum von-Mises equivalent stress value, in the case of type 1 OPLL was 8.43% for C4 laminectomy, 20.95% for C5 laminectomy, and 64.20% for C4–C5 laminectomy. In type 2 OPLL, the decompression rate was 3.08% for C4 laminectomy, 53.16% for C5 laminectomy, and 65.73% for C4–C5 laminectomy. In type 3 OPLL, the decompression rate was −16.58% for C4 laminectomy, 8.67% for C5 laminectomy, and 58.86% for C4–C5 laminectomy.

The decompression rate of the average von-Mises equivalent stress value, in the case of type 1 OPLL was 7.35% for C4 laminectomy, 16.67% for C5 laminectomy, and 43.87% for C4–C5 laminectomy. In type 2 OPLL, the decompression rate was 28.02% for C4 laminectomy, 47.51% for C5 laminectomy, and 61.60% for C4–C5 laminectomy. In type 3 OPLL, the decompression rate was 28.90% for C4 laminectomy, 38.68% for C5 laminectomy, and 57.72% for C4–C5 laminectomy.

### 3.3. Flexion Laminectomy Simulation

The results of the flexion simulation and decompression are shown in Figure 10 and Table 7, Table 8 and Table 9.

The % expressed below is the decompression rate when each laminectomy is applied based on the non-surgery model for each type of OPLL. The −(minus) value indicates that the stress acting on the spinal cord increased.

The decompression rate of the maximum principal stress value, in the case of type 1 OPLL was 18.42% for C4 laminectomy, −17.02% for C5 laminectomy, and 25.26% for C4–C5 laminectomy. In type 2 OPLL, the decompression rate was 16.65% for C4 laminectomy, 15.26% for C5 laminectomy, and 32.67% for C4–C5 laminectomy. In type 3 OPLL, the decompression rate was 32.34% for C4 laminectomy, 18.35% for C5 laminectomy, and 54.79% for C4–C5 laminectomy.

The decompression rate of the minimum principal stress value, in the case of type 1 OPLL was 29.07% for C4 laminectomy, −26.45% for C5 laminectomy, and 40.73% for C4–C5 laminectomy. In type 2 OPLL, the decompression rate was −33.35% for C4 laminectomy, 5.86% for C5 laminectomy, and 49.33% for C4–C5 laminectomy. In type 3 OPLL, the decompression rate was 9.78% for C4 laminectomy, −12.17% for C5 laminectomy, and 29.60% for C4–C5 laminectomy.

The decompression rate of the maximum von Mises equivalent stress value, in the case of type 1 was −19.04% for C4 laminectomy, −22.99% for C5 laminectomy, and 25.34% for C4–C5 laminectomy. In type 2 OPLL, the decompression rate was −22.53% for C4 laminectomy, 18.40% for C5 laminectomy, and 47.10% for C4–C5 laminectomy. In type 3 OPLL, the decompression rate was −75.14% for C4 laminectomy, −14.61% for C5 laminectomy, and 27.78% for C4–C5 laminectomy.

The decompression rate of the average von Mises equivalent stress value, in the case of type 1 OPLL was 21.53% for C4 laminectomy, −17.20% for C5 laminectomy, and 25.28% for C4–C5 laminectomy. In type 2 OPLL, the decompression rate was 16.08% for C4 laminectomy, 14.58% for C5 laminectomy, and 35.35% for C4–C5 laminectomy. In type 3 OPLL, the decompression rate was 34.08% for C4 laminectomy, 13.60% for C5 laminectomy, and 40.10% for C4–C5 laminectomy.

### 3.4. Extension Laminectomy Simulation

The results of the extension simulation and decompression are shown in Figure 11 and Table 10, Table 11 and Table 12.

The % expressed below is the decompression rate when each laminectomy is applied based on the non-surgery model for each type of OPLL. The −(minus) value indicates that the stress acting on the spinal cord increased.

The decompression rate of the maximum principal stress value, in the case of type 1 OPLL was 22.64% for C4 laminectomy, 3.90% for C5 laminectomy, and 30.45% for C4–C5 laminectomy. In type 2 OPLL, the decompression rate was 13.74% for C4 laminectomy, 0.53% for C5 laminectomy, and −6.36% for C4–C5 laminectomy. In type 3 OPLL, the decompression rate was 35.71% for C4 laminectomy, 12.54% for C5 laminectomy, and −39.13% for C4–C5 laminectomy.

The decompression rate of the minimum principal stress value, in the case of type 1 OPLL was 32.96% for C4 laminectomy, 36.38% for C5 laminectomy, and 84.78% for C4–C5 laminectomy. In type 2 OPLL, the decompression rate was 32.02% for C4 laminectomy, 58.10% for C5 laminectomy, and 85.85% for C4–C5 laminectomy. In type 3 OPLL, the decompression rate was 8.56% for C4 laminectomy, −39.07% for C5 laminectomy, and 47.76% for C4–C5 laminectomy.

The decompression rate of the maximum von Mises equivalent stress value, in the case of type 1 OPLL was 38.36% for C4 laminectomy, 42.85% for C5 laminectomy, and 79.98% for C4–C5 laminectomy. In type 2 OPLL, the decompression rate was 28.09% for C4 laminectomy, 71.76% for C5 laminectomy, and 78.21% for C4–C5 laminectomy. In type 3 OPLL, the decompression rate was 13.24% for C4 laminectomy, −18.78% for C5 laminectomy, and 41.88% for C4–C5 laminectomy.

The decompression rate of the average von Mises equivalent stress value, in the case of type 1 was 18.57% for C4 laminectomy, 14.62% for C5 laminectomy, and 41.23% for C4–C5 laminectomy. In type 2 OPLL, the decompression rate was 14.73% for C4 laminectomy, 12.97% for C5 laminectomy, and 25.92% for C4–C5 laminectomy. In type 3 OPLL, the decompression rate was 24.23% for C4 laminectomy, 4.66% for C5 laminectomy, and 49.22% for C4–C5 laminectomy.

## 4. Discussion

In previous studies on the cervical spine, ligament models were generally composed of beams [24,25]. However, in this study, the LF and CL were constructed as a solid-type 3D model to analyze all structures that could compress the spinal cord. Compression of the spinal cord due to excessive buckling of the LF may occur [26,27]. When the spinal cord is pushed back by the OPLL, the ROM of the spinal cord is limited, as in an actual human body. In addition, cerebrospinal fluid (CSF) was excluded during the construction of the spinal cord model in this study, and the physical property value of the dura mater was applied to the cord surface using the surface coating function in the ANSYS program. This is configured accordingly as when the spinal cord is pressed by the OPLL, the CSF is pushed out, and the dura mater comes into contact with the cord and transfers the pressure. Herein, the OPLL model was composed of three types showing a large number of frequencies. The constructed OPLL model was then inserted into the C4–C5 segment. This is due to the canal being small and having a wide ROM; therefore, it has a large impact on the spinal cord [28].

The validation results of the intact cervical spine model were confirmed to be within the range of the cadaveric experimental data by Panjabi, which showed that the normal cervical model constructed in this study is suitable for simulating human behaviors.

All simulation results were confirmed to be lower than the tensile strength values of the spinal cord and ligaments, indicating that there was no failure of the spinal cord and ligaments in the simulation [29,30,31].

Based on the results of the neutral simulation, it was confirmed that the decompression applied to C5 was better than that applied to C4 when laminectomy was performed on a single segment. In the case of C4 laminectomy of type 3 OPLL, the maximum equivalent stress increased. The neutral simulation confirmed that the decompression was good when the C4–C5 laminectomy was performed. However, as this also showed an average decompression rate of 60%, it was confirmed that the residual stress in the spinal cord remained.

Flexion and extension laminectomy simulations were performed by extracting the deformed spinal cord from the neutral-state simulation. As only the deformed spinal cord was extracted, the simulation proceeded from the zero base, which did not have any stress acting on the cord. This was analyzed for the decompression of laminectomy during flexion and extension, considering that the stress acting on the spinal cord in the neutral state is very small as the spinal cord has deformed over time.

Based on the results of the flexion laminectomy simulation, it was confirmed that the maximum equivalent stress increased after C4 laminectomy of type 1 OPLL and that all stresses increased after C5 laminectomy. The minimum principal and maximum equivalent stresses increased after C4 laminectomy of type 2 OPLL. Finally, the maximum equivalent stress increased after the C4 laminectomy of type 3 OPLL.

Based on the results of the extension laminectomy simulation, it was confirmed that the maximum principal stress increased after C5 and C4–C5 laminectomies of type 1 OPLL. The maximum principal stress increased after C4–C5 laminectomy of type 2 OPLL. Finally, the minimum principal and maximum equivalent stresses after C5 laminectomy and the maximum principal stress after C4–C5 laminectomy of type 3 OPLL increased.

Based on the results of the neutral simulation, decompression was considered good when laminectomy was performed; however, it was confirmed that the stress value increased during flexion and extension. Taken together, the surgical prognosis may be poor owing to residual stress and increasing stress when laminectomy is performed on patients with focal OPLL.

## 5. Conclusions

The results for the neutral state were good. Therefore, if emergency decompression is required, it may be possible to perform posterior decompression first. However, based on the flexion and extension simulation results, posterior decompression alone has a poor prognosis in cases of focal OPLL; therefore, an additional anterior approach should be considered as early as possible.

## Figures and Tables

**Figure 1 bioengineering-09-00519-f001:**
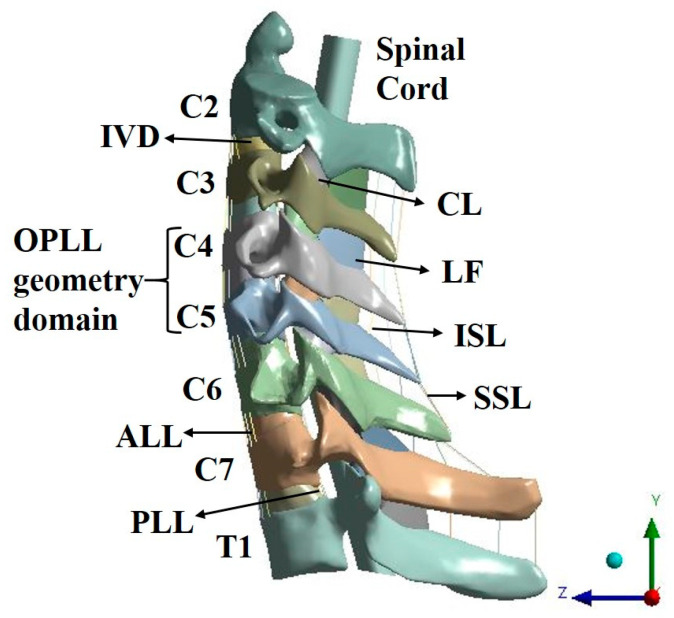
Three-dimensional cervical spine model.

**Figure 2 bioengineering-09-00519-f002:**
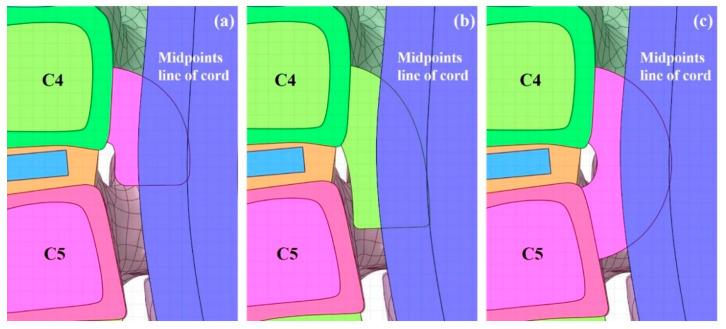
Three-dimensional OPLL model: (**a**) type 1 OPLL: covered disc; (**b**) type 2 OPLL: covered vertebra; and (**c**) type 3 OPLL: connected vertebra.

**Figure 3 bioengineering-09-00519-f003:**
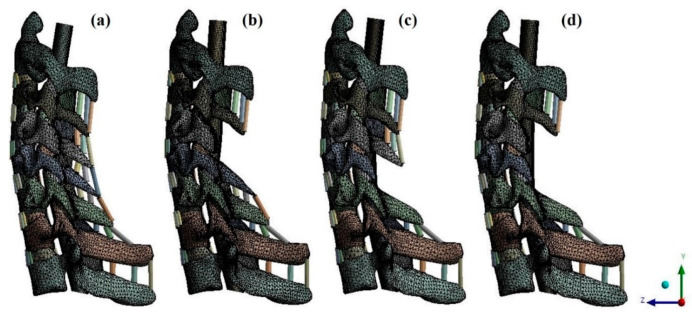
Finite element model of the OPLL cervical spine model: (**a**) non-surgery model; (**b**) C4 laminectomy model; (**c**) C5 laminectomy model; and (**d**) C4–C5 laminectomy model. OPLL, ossification of the posterior longitudinal ligament.

**Figure 4 bioengineering-09-00519-f004:**
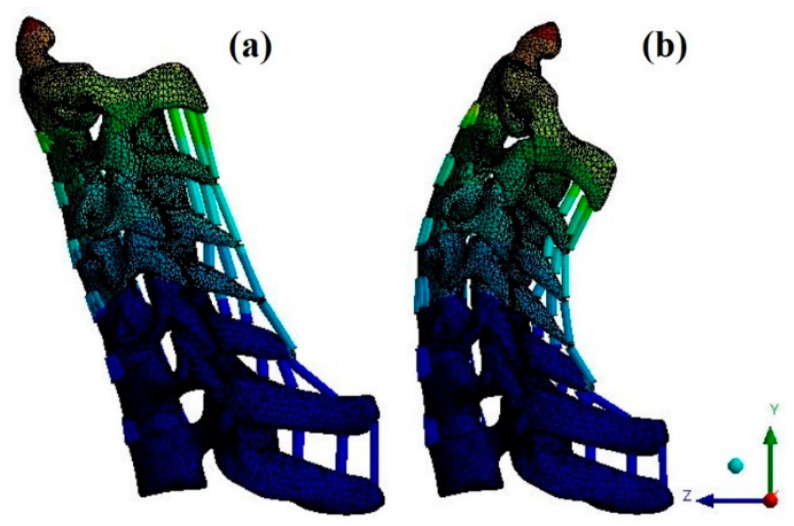
Finite element model: (**a**) flexion; and (**b**) extension.

**Figure 5 bioengineering-09-00519-f005:**
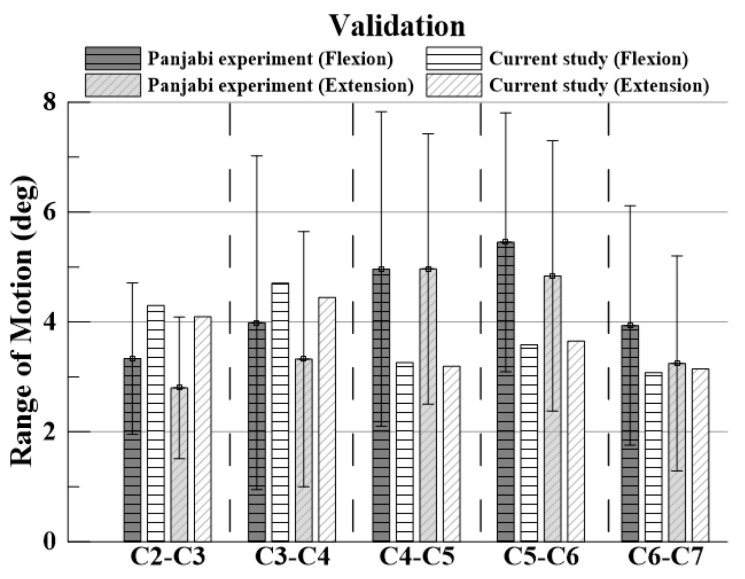
Validation based on Panjabi’s experimental data.

**Figure 6 bioengineering-09-00519-f006:**
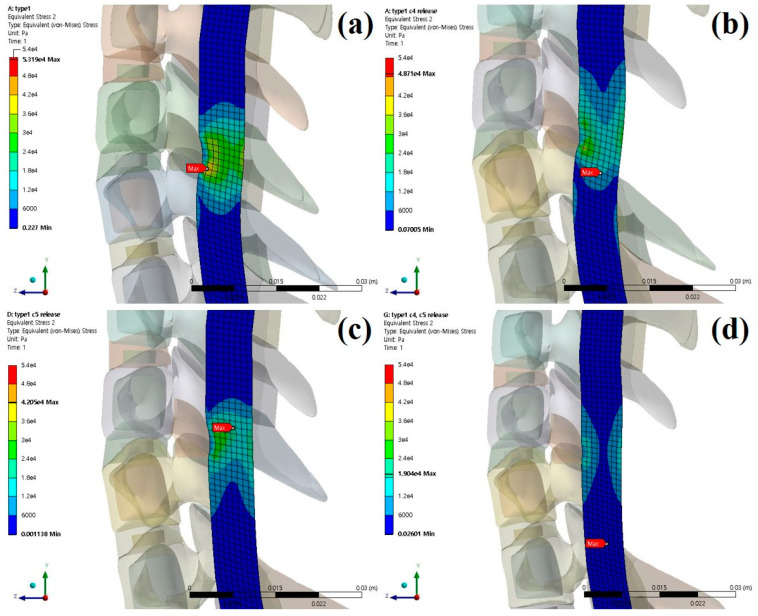
Neutral laminectomy simulation type 1 OPLL: (**a**) non-surgery model equivalent stresses (Pa); (**b**) C4 laminectomy model equivalent stresses (Pa); (**c**) C5 laminectomy model equivalent stresses (Pa); and (**d**) C4–C5 laminectomy model equivalent stresses (Pa).

**Figure 7 bioengineering-09-00519-f007:**
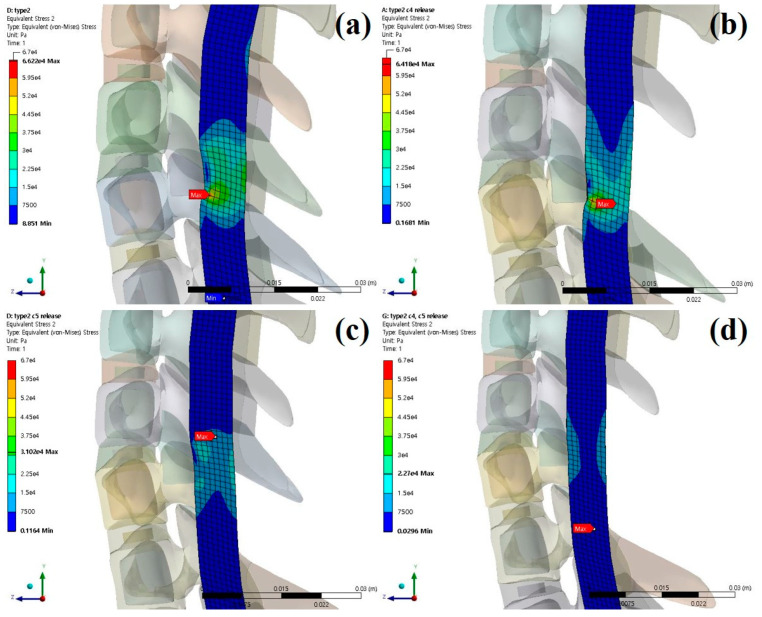
Neutral laminectomy simulation type 2 OPLL: (**a**) non-surgery model equivalent stresses (Pa); (**b**) C4 laminectomy model equivalent stresses (Pa); (**c**) C5 laminectomy model equivalent stresses (Pa); and (**d**) C4–C5 laminectomy model equivalent stresses (Pa).

**Figure 8 bioengineering-09-00519-f008:**
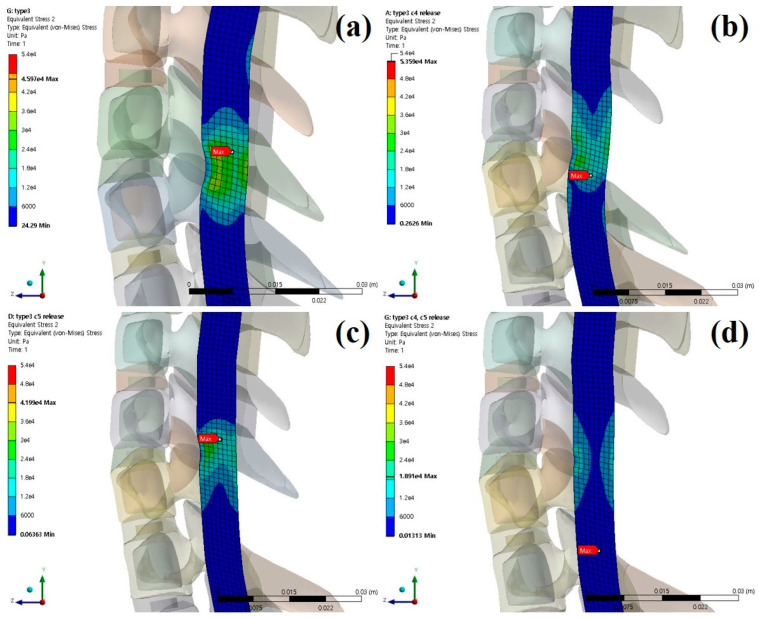
Neutral laminectomy simulation type 3 OPLL: (**a**) non-surgery model equivalent stresses (Pa); (**b**) C4 laminectomy model equivalent stresses (Pa); (**c**) C5 laminectomy model equivalent stresses (Pa); and (**d**) C4–C5 laminectomy model equivalent stresses (Pa).

**Figure 9 bioengineering-09-00519-f009:**
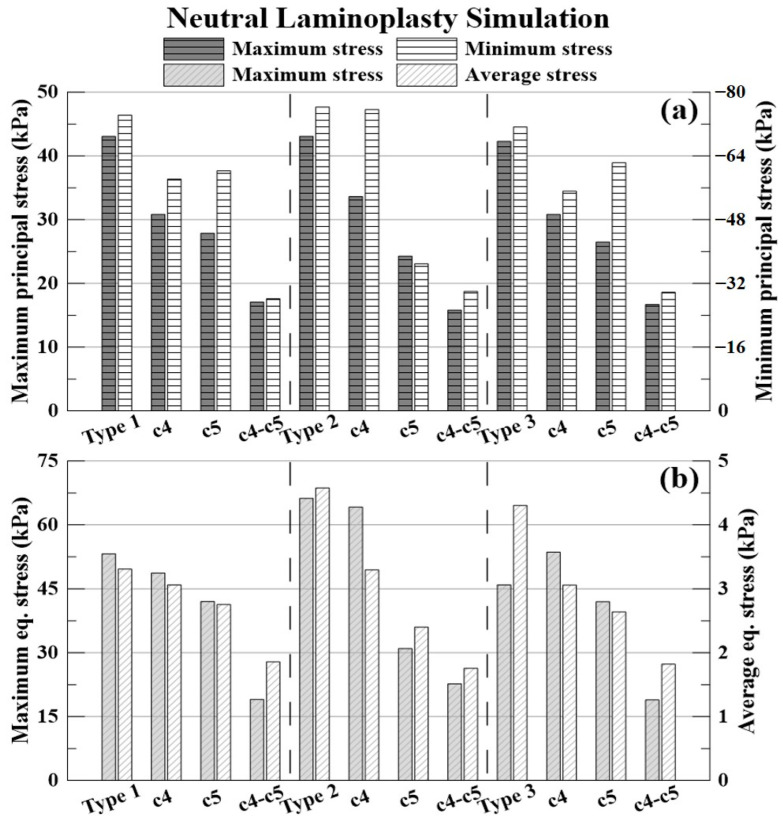
Neutral laminectomy simulation: (**a**) maximum and minimum principal stresses (kPa); and (**b**) maximum and average equivalent stresses (kPa).

**Figure 10 bioengineering-09-00519-f010:**
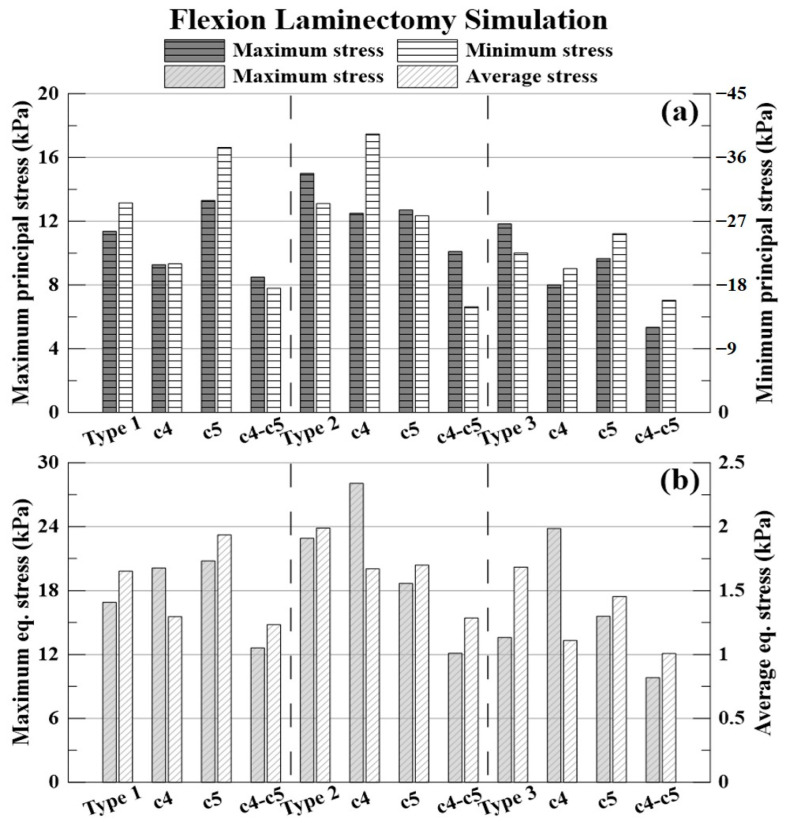
Flexion laminectomy simulation: (**a**) maximum and minimum principal stresses (kPa); and (**b**) maximum and average equivalent stresses (kPa).

**Figure 11 bioengineering-09-00519-f011:**
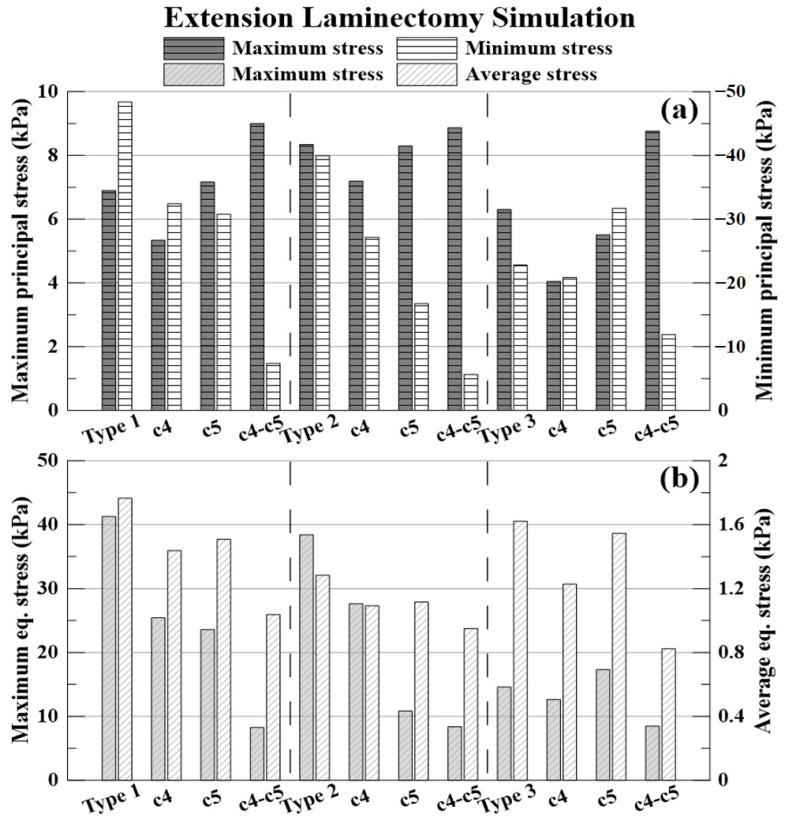
Extension laminectomy simulation: (**a**) maximum and minimum principal stresses (kPa); and (**b**) maximum and average equivalent stresses (kPa).

**Table 1 bioengineering-09-00519-t001:** Material properties of the cervical spine model.

Material		Young’s Modulus (MPa)	Poisson’s Ratio	Reference
Vertebra	Cortical bone	12,000	0.29	[11]
	Cancellous bone	450		
IVD	Nucleus	Mooney-Rivlin Model C10 = 0.12, C01 = 0.09, D1 = 0		[12]
	Annulus	Mooney-Rivlin Model C10 = 0.133, C01 = 0.0333, D1 = 0.6		
Spinal Cord	Dura mater	5	0.49	[13]
	Cord	0.26		
Ligament	ALL	10	0.3	[11]
	PLL	10		
	ISL	1.5		
	SSL	1.5		
	CL	10		
	LF	1.5		

**Table 2 bioengineering-09-00519-t002:** Cross-sectional area of spinal ligaments.

Ligament	Cross-Sectional Area (mm^2^)	Reference
ALL	10.6	[14]
PLL	1.6	
ISL	12	
SSL	6	

**Table 3 bioengineering-09-00519-t003:** Type and number of elements on finite element model.

Material		Type	No. of Elements
Vertebra	Cortical bone	10-node tetrahedral element	128,367
	Cancellous bone		19,247
IVD	Nucleus		5392
	Annulus		9902
Spinal Cord	Dura mater	Surface-coating	
	Cord	20-node hexahedral element	16,541
OPLL	Type 1		14,095
	Type 2		17,773
	Type 3		21,114
Ligament	ALL	Tension-only beam	30
	PLL		30
	ISL		12
	SSL		12
	CL	10-node tetrahedral element	2319
	LF		12,861

**Table 4 bioengineering-09-00519-t004:** Neutral laminectomy simulation spinal cord stresses: type 1 OPLL.

Unit: kPa	Type 1 OPLL	C4 Laminectomy	C5 Laminectomy	C4–C5 Laminectomy
Max. principal	43.05	30.82	27.87	17.06
Decompression rate	0.00%	28.40%	35.26%	60.37%
Min. principal	−74.24	−58.21	−60.30	−28.16
Decompression rate	0.00%	21.60%	18.78%	62.07%
Max. equivalent	53.19	48.71	42.05	19.04
Decompression rate	0.00%	8.43%	20.95%	64.20%
Avg. equivalent	3.31	3.06	2.76	1.86
Decompression rate	0.00%	7.35%	16.67%	43.87%

**Table 5 bioengineering-09-00519-t005:** Neutral laminectomy simulation spinal cord stresses: type 2 OPLL.

Unit: kPa	Type 2 OPLL	C4 Laminectomy	C5 Laminectomy	C4–C5 Laminectomy
Max. principal	43.07	33.63	24.28	15.80
Decompression rate	0.00%	21.93%	43.63%	63.32%
Min. principal	−76.29	−75.64	−36.93	−29.99
Decompression rate	0.00%	0.85%	51.59%	60.69%
Max. equivalent	66.22	64.18	31.02	22.70
Decompression rate	0.00%	3.08%	53.16%	65.73%
Avg. equivalent	4.58	3.29	2.40	1.76
Decompression rate	0.00%	28.02%	47.51%	61.60%

**Table 6 bioengineering-09-00519-t006:** Neutral laminectomy simulation spinal cord stresses: type 3 OPLL.

Unit: kPa	Type 3 OPLL	C4 Laminectomy	C5 Laminectomy	C4–C5 Laminectomy
Max. principal	42.29	30.82	26.48	16.69
Decompression rate	0.00%	27.12%	37.38%	60.54%
Min. principal	−71.29	−55.14	−62.30	−29.76
Decompression rate	0.00%	22.65%	12.61%	58.26%
Max. equivalent	45.97	53.59	41.99	18.91
Decompression rate	0.00%	−16.58%	8.67%	58.86%
Avg. equivalent	4.31	3.06	2.64	1.82
Decompression rate	0.00%	28.90%	38.68%	57.72%

**Table 7 bioengineering-09-00519-t007:** Flexion laminectomy simulation spinal cord stresses: type 1 OPLL.

Unit: kPa	Type 1 OPLL	C4 Laminectomy	C5 Laminectomy	C4–C5 Laminectomy
Max. principal	11.37	9.27	13.30	8.50
Decompression rate	0.00%	18.42%	−17.02%	25.26%
Min. principal	−29.60	−21.00	−37.43	−17.55
Decompression rate	0.00%	29.07%	−26.45%	40.73%
Max. equivalent	16.89	20.11	20.77	12.61
Decompression rate	0.00%	−19.04%	−22.99%	25.34%
Avg. equivalent	1.65	1.30	1.94	1.23
Decompression rate	0.00%	21.53%	−17.20%	25.28%

**Table 8 bioengineering-09-00519-t008:** Flexion laminectomy simulation spinal cord stresses: type 2 OPLL.

Unit: kPa	Type 2 OPLL	C4 Laminectomy	C5 Laminectomy	C4–C5 Laminectomy
Max. principal	15.00	12.50	12.71	10.10
Decompression rate	0.00	16.65%	15.26%	32.67%
Min. principal	−29.49	−39.33	−27.77	−14.94
Decompression rate	0.00%	−33.35%	5.86%	49.33%
Max. equivalent	22.89	28.05	18.68	12.11
Decompression rate	0.00%	−22.53%	18.40%	47.10%
Avg. equivalent	1.99	1.67	1.70	1.29
Decompression rate	0.00%	16.08%	14.58%	35.35%

**Table 9 bioengineering-09-00519-t009:** Flexion laminectomy simulation spinal cord stresses: type 3 OPLL.

Unit: kPa	Type 3 OPLL	C4 Laminectomy	C5 Laminectomy	C4–C5 Laminectomy
Max. principal	11.83	8.00	9.66	5.35
Decompression rate	0.00%	32.34%	18.35%	54.79%
Min. principal	−22.52	−20.32	−25.26	−15.86
Decompression rate	0.00%	9.78%	−12.17%	29.60%
Max. equivalent	13.60	23.82	15.59	9.82
Decompression rate	0.00%	−75.14%	−14.61%	27.78%
Avg. equivalent	1.68	1.11	1.45	1.01
Decompression rate	0.00%	34.08%	13.60%	40.10%

**Table 10 bioengineering-09-00519-t010:** Extension laminectomy simulation spinal cord stresses: type 1 OPLL.

Unit: kPa	Type 1 OPLL	C4 Laminectomy	C5 Laminectomy	C4–C5 Laminectomy
Max. principal	6.90	5.33	7.17	9.00
Decompression rate	0.00%	22.64%	−3.90%	−30.45%
Min. principal	−48.39	−32.44	−30.78	−7.36
Decompression rate	0.00%	32.96%	36.38%	84.78%
Max. equivalent	41.27	25.44	23.58	8.26
Decompression rate	0.00%	38.36%	42.85%	79.98%
Avg. equivalent	1.76	1.44	1.51	1.04
Decompression rate	0.00%	18.57%	14.62%	41.23%

**Table 11 bioengineering-09-00519-t011:** Extension laminectomy simulation spinal cord stresses: type 2 OPLL.

Unit: kPa	Type 2 OPLL	C4 Laminectomy	C5 Laminectomy	C4–C5 Laminectomy
Max. principal	8.34	7.19	8.30	8.87
Decompression rate	0.00%	13.74%	0.53%	−6.36%
Min. principal	−39.96	−27.17	−16.74	−5.65
Decompression rate	0.00%	32.02%	58.10%	85.85%
Max. equivalent	38.38	27.60	10.84	8.36
Decompression rate	0.00%	28.09%	71.76%	78.21%
Avg. equivalent	1.28	1.09	1.12	0.95
Decompression rate	0.00%	14.73%	12.97%	25.92%

**Table 12 bioengineering-09-00519-t012:** Extension laminectomy simulation spinal cord stresses: type 3 OPLL.

Unit: kPa	Type 3 OPLL	C4 Laminectomy	C5 Laminectomy	C4–C5 Laminectomy
Max. principal	6.30	4.05	5.51	8.76
Decompression rate	0.00%	35.71%	12.54%	−39.13%
Min. principal	−22.81	−20.86	−31.72	−11.92
Decompression rate	0.00%	8.56%	−39.07%	47.76%
Max. equivalent	14.57	12.64	17.31	8.47
Decompression rate	0.00%	13.24%	−18.78%	41.88%
Avg. equivalent	1.62	1.23	1.55	0.82
Decompression rate	0.00%	24.23%	4.66%	49.22%

## Data Availability

Data sharing not applicable.

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
