# Peer review of "Stress Distribution on Spinal Cord According to Type of Laminectomy for Large Focal Cervical Ossification of Posterior Longitudinal Ligament Based on Finite Element Method"

_bioengineering, 2022, doi:10.3390/bioengineering9100519_

Round 1

Reviewer 1 Report

This paper described stress distribution on the spinal cord after laminectomy for large cervical ossification of the posterior longitudinal ligament by the finite element method. There are some ambiguities and questions in this paper.

1.      In this model, the capsular ligament (CL) was constructed as a solid-type 3D model (line 215). Under the condition that the CL is fixed, can the cervical spine move? What were the contact conditions applied to the facet joint?

2.      The description and definition of Type 1 OPLL, Type 2 OPLL, and Type 3 OPLL were not given.

3.      Has the spinal cord been shifted posteriorly after laminectomy in the model? If so, what is the distance? There can be deformation of the spinal cord under compression and after laminectomy. In such conditions, the stress distribution in the spinal cord differs between the ventral and dorsal sides. How were the intramedullary stresses determined? If it was expressed as an average over a certain range, the range should be clearly indicated. There needs to be a schema showing the stress distribution in the spinal cord.

4.      The K-line is a line connecting the midpoints of the cervical spinal canal at C2 and C7 on a neutral cervical lateral radiograph, not the line by connecting the midpoints of the spinal cord on the lateral radiograph of the cervical spine (Line 82). The authors must correct the sentence and figure.

Author Response

Thank you for taking the time of the judges, and we will answer your questions as follows.

Reviewer 2 Report

The article presents a comparison of stress when suffering posterior longitudinal ligament ossification when a laminectomy has been performed. Several scenarios were simulated: when the operation is performed on the C4 vertebra, when the C5 vertebra is operated and when a procedure is performed on both vertebrae. In addition, there are three types of models:

 1) when the K line covers the disc

2) when the K line covers the vertebra

3) when the K line is tangent (connects) to the vertebra

4) when the k-line is tangent (connects) to the vertebra, this mentioned above gives 12 case studies.

 However, some issues are not specified as the considerations that were made for the material. For example, in table 1, the Mooney-Rivlin coefficients are indicated. If this constitutive model was not used for the simulation, omit those values. In addition, a better description of the mesh and the type of element used for the simulation is missing.

 A mesh sensitivity analysis is recommended for the model validation section (109).

 Other recommendations:

 - Indicate why the weight of the person is not considered.

- Explain why a moment of 1 Nm is applied (line 75).

- Include a brief explanation of the percentages obtained in the results section.

- Include an analysis of the deformation of the element.

- In lines 128, 158, and 188, the term "pressure" is used when it would be more appropriate to use the word "stress"

- Use a table comparing the results obtained and those of the other researchers mentioned in line 233.

 The introduction is well done but contains references that are not current (2005, 1960, 2006, 2002, 2003, 1994).

 It is pertinent to include a better description of how the spine model was generated, the considerations taken about the material (linear, continuous, isotropic, etc.), a description of the mesh used for the simulation, and justifying the loads that are being applied.

The implementation of tables is necessary to show the results better. They can be simplified to show the maximum stresses (Von-Mises) since, based on the failure theory used (maximum distortion energy), the principal stresses are related to the Von-Mises stress. It is recommended to add deformation analysis. It is confusing that the tables show the stresses in their respective units and appear as percentages in the wording.

The list of bibliographical references is not appropriate, current, and following the journal's guidelines. It even has references from 1960.

Adding an image showing a diagram describing the methodology used is recommended.

A description of how the finite element model was obtained is missing, as well as the type of mesh used. Therefore, a mesh sensitivity analysis is recommended.

Comparing the results obtained with research that contains experimental tests is recommended.

Explain the difference between the proposed research and the following publication, "Numerical Evaluation of Spinal Stability after Posterior Spinal Fusion with Various Fixation Segments and Screw Types in Patients with Osteoporotic Thoracolumbar Burst Fracture Using Finite Element Analysis".

Explain in a better way the graphs. For example, talking about a stress decrease or increase is confusing without explaining what is being compared.

Use a more extensive range concerning the stresses reported in Figure 7 so that the size does not appear to limit the graph.

Author Response

(The authors gave the same response as above.)

Round 2

Reviewer 1 Report

NA